# Nontuberculous Mycobacterial Lung Disease in the Patients with Cystic Fibrosis—A Challenging Diagnostic Problem

**DOI:** 10.3390/diagnostics12071514

**Published:** 2022-06-21

**Authors:** Dorota Wyrostkiewicz, Lucyna Opoka, Dorota Filipczak, Ewa Jankowska, Wojciech Skorupa, Ewa Augustynowicz-Kopeć, Monika Szturmowicz

**Affiliations:** 1Ist Department of Lung Diseases, National Tuberculosis and Lung Diseases Research Institute, 01-138 Warsaw, Poland; esrubka@gmail.com (E.J.); w.skorupa@igichp.edu.pl (W.S.); monika.szturmowicz@gmail.com (M.S.); 2Department of Radiology, National Tuberculosis and Lung Diseases Research Institute, 01-138 Warsaw, Poland; lucyna.opoka@gmail.com; 3Department of Microbiology, National Tuberculosis and Lung Diseases Research Institute, 01-138 Warsaw, Poland; d.filipczak@igichp.edu.pl (D.F.); e.kopec@igichp.edu.pl (E.A.-K.)

**Keywords:** nontuberculous mycobacteria, cystic fibrosis, chest computed tomography, respiratory infection

## Abstract

Background: Cystic fibrosis (CF) is an autosomal, recessive genetic disorder, caused by a mutation in the cystic fibrosis transmembrane conductance receptor regulator (CFTR) gene. Dysregulated mucous production, and decreased bronchial mucociliary clearance, results in increased susceptibility to bacterial and fungal infections. Recently, nontuberculous mycobacteria (NTM) infections were identified as an emerging clinical problem in CF patients. Aim: The aim of the present study was to assess the frequency of NTM isolations in CF patients hospitalized in the pulmonary department, serving as a hospital CF center, and to describe challenges concerning the recognition of NTMLD (nontuberculous mycobacterial lung disease) in those patients. Methods: Consecutive CF patients, who were hospitalized due to pulmonary exacerbations (PEX), in a single CF center, between 2010 and 2020, were retrospectively assessed for the presence of NTM in respiratory specimens. Clinical and radiological data were retrospectively reviewed. Results: Positive respiratory specimen cultures for NTM were obtained in 11 out of 151 patients (7%), mean age—35.7 years, mean BMI—20.2 kg/m^2^, mean FEV1—58.6% pred. Cultures and phenotyping revealed the presence of *Mycobacterium avium* (*M. avium*)—in six patients, *Mycobacterium chimaera* (*M. chimaera*) in two, *Mycobacterium kansasii* (*M. kansasii*)—in one, *Mycobacterium abscessus* (*M. abscessus*)—in one, *Mycobacterium lentifavum* (*M. lentiflavum*)—in one. Simultaneously, respiratory cultures were positive for fungi in 91% of patients: *Candida albicans* (*C. albicans*)—in 82%, *Aspergillus fumigatus (A. fumigatus*)—in 45%. Clinical signs of NTMLD were non—specific, chest CT indicated NTMLD in five patients only. Conclusion: Due to non-specific clinical presentation, frequent sputum cultures for NTM and analysis of serial chest CT examinations are crucial for NTMLD recognition in CF patients. Further studies concerning the predictive role of fungal pathogens for NTMLD development in CF patients are needed.

## 1. Introduction

Cystic fibrosis (CF) is an inherited, autosomal, recessive genetic disorder caused by mutation in the cystic fibrosis transmembrane conductance receptor regulator (CFTR) gene [1]. It affects 1 in 2000–3000 newborns [2]. The CFTR gene is responsible for the regulation of the chloride channel, located at the apical surface of various epithelial cells [1,3]. CFTR mutation results in the dysregulation of the chloride ions flow, increased water absorption and secretion condensation. In the respiratory tract, the presence of thick mucus and impaired bronchial mucociliary clearance cause increased susceptibility to bacterial and fungal colonization [1]. As the eradication of pathogens is disturbed, chronic colonization, with temporary exacerbations of inflammatory bronchitis is observed.

Recently, infections with nontuberculous mycobacteria (NTM) were identified as an emerging clinical problem in CF patients [4,5]. Positive respiratory cultures for NTM were obtained in CF patients 400 times more frequently compared to the general population [6]. The most frustrating scenario is the possibility of direct or indirect person-to-person transmission of NTM in CF centers, especially following *M. abscessus* colonization.

The recognition of lung disease caused by NTM (NTMLD) is based on clinical, radiological and microbiological criteria. Nevertheless, in CF patients NTMLD is very difficult to diagnose due to many similarities in clinical and radiological presentation between pulmonary exacerbations caused by bacterial pathogens and those caused by NTM.

## 2. Aim

The aim of the present study was to assess the frequency of NTM isolations in CF patients hospitalized in the pulmonary department, serving as a hospital CF center, and to describe challenges concerning the recognition of NTMLD in those patients.

## 3. Material and Methods

Cystic fibrosis patients, who were hospitalized due to pulmonary exacerbations (PEX), in the 1st Department of Lung Diseases National Tuberculosis and Lung Diseases Research Institute, between 2010 and 2020, were retrospectively assessed for the presence of NTM in respiratory specimens.

CF was recognized based on clinical and radiological symptoms, high level of chloride in sweat tests and finally confirmed with the presence of CFTR mutations.

Clinical data registered in the hospital database at the time of NTM isolation were analyzed. Chest CT scans performed at the time of NTM isolation were reviewed by the two pulmonologists and one radiology specialist and compared to previous radiologic documentation of the patients.

The sputum or bronchial washings obtained during fiberoptic bronchoscopy were decontaminated with sodium hydrochloride and N-acetyl-L-cysteine. Smears for acid-fast bacilli (AFB) were stained with auramine fluorochrome. Fluorochrome-positive specimens were confirmed by the Ziehl–Neelsen method. Gene Xpert MTB/Ultra Cepheid test was performed in the case of AFB positivity. The strains were cultured on a solid medium (egg-based Lowenstein–Jensen) and in the automated system MGIT (Becton Dickinson, Franklin Lakes, NJ, USA).

Identification of NTM species was performed with the GenoType CM test (Hain Lifescience), versions 1.0 and 2.0. *M. chimera* was identified with the GenoType NTM-DR test, based on 23S rRNA gene polymorphism. The principles of the procedure were described previously [7].

The project was accepted by the Institutional Ethical Committee (No. 9/2015). 

## 4. Results

One hundred and fifty-one CF patients have been hospitalized between 2010 and 2020, due to PEX. Positive respiratory specimen cultures for NTM were obtained in 11 patients (7%). The group consisted of nine females and two males, mean age—35.7 years (SD 9.96 years). Non-tuberculous mycobacterial lung disease (NTMLD) was diagnosed in eight patients, and respiratory system colonization with NTM—in three. Anti-mycobacterial therapy was administered in six patients with NTMLD, one patient refused treatment; in one case the treatment was delayed due to large clinical improvement in the course of PEX therapy. The patients were treated according to European Society for Cystic Fibrosis recommendations [6].

### 4.1. Clinical Data

Mean BMI was—20.2 kg/m^2^ (SD 2.78 kg/m^2^). Mean spirometry values were: FVC—73.8% (SD 16.7%) predicted, FEV1—58.6% (SD 20.6%) predicted, FEV1/FVC—70.1 (SD 14.33).

Allergic bronchopulmonary aspergillosis (ABPA) was diagnosed in five patients (46%), three of them had been treated with prednisone at the time of NTM isolation. All of the patients were non-smokers.

Type 1 diabetes mellitus was diagnosed in three patients (27%), gastroesophageal reflux—in one, and hypothyreosis—in one.

On admission, increased expectoration of purulent sputum was present in 11 patients (100%), decrease in exercise tolerance—in eight patients (73%), loss of appetite—in eight patients (73%), and increased body temperature—in six (55%), hemoptysis—in three (27%).

### 4.2. Microbiological Analysis

#### 4.2.1. NTM

AFB smears were positive in four (36%) patients. In all of them, Gene Xpert MTB tests were negative. Cultures and phenotyping revealed the presence of: *M. avium*—in six patients, *M. chimaera*—in two, *M. kansasii*—in one, *M. abscessus*—in one, *M. lentiflavum*—in one (Table 1). 

All of the patients fulfilled the microbiological criteria of NTMLD diagnosis (at least two positive sputum samples or one positive bronchial washing sample). 

#### 4.2.2. Other Pathogens

At the time of NTM isolation seven patients (63%) were colonized with *Pseudomonas aeruginosa (P. aeruginosa)*, six (55%)—with *Staphylococcus aureus (S. aureus)*, and four patients (36%)—both *P. aeruginosa* and *S. aureus* (Table 1).

Respiratory cultures were positive for fungi—in 10 (91%) patients, *Aspergillus fumigatus* (*A. fumigatus*) was isolated in 5 (45%), *Candida albicans* (*C. albicans*)—in 9 (82%), and both types of fungi—in 5 (45%) patients (Table 1).

### 4.3. Radiological Data

Chest CT revealed bilateral bronchiectasis, localized predominantly in the upper lung lobes and thickened bronchial walls with the signs of mucus plugging in the bronchial lumen, in all patients. In three patients—enlarged mediastinal lymph nodes, and in two—air trapping was described. Partial lung cirrhosis was noted in one patient.

In five out of 11 patients, chest CT was indicative of NTMLD. New nodular infiltrations with cavitation were described in three patients and new areas of centrilobular nodules in the middle lobe and lingua—in two patients (Figure 1 and Figure 2). In the remaining six patients, the radiologic appearance of chest CT was not suggestive of NTMLD. 

## 5. Discussion

NTM are increasingly identified in respiratory specimens all over the world. In the US the estimated number of NTM pulmonary cases increased two-fold between 2010 and 2014 [8]. The growing number of NTM isolates is probably caused by the aging of populations and the increasing number of patients with chronic lung diseases predisposing to NTM colonization [9].

In CF patients, the estimated risk of NTM pulmonary infection is approximately 1000 times higher compared to the healthy population [1].

At our center, NTM was isolated in 7% of 151 hospitalized CF patients, during 10 years of follow-up. The prevalence of positive NTM isolations in other European CF centers was 4–10% [10,11,12]. A lower prevalence of NTM infections was noted among patients less than 16 years of age (1.3% in 2010 and 3.8% in 2015) [13]. In the US CF centers, the annual prevalence of positive NTM isolates was 2–14% and the frequency of positive isolations in CF increased from 11% per year in 2010 to 13.4% in 2014 [6,14].

The principles of NTMLD recognition have been summarized recently [15]. The diagnosis is based on clinical, radiological and microbiological data (Table 2).

Nevertheless, the problem of NTMLD diagnosis in CF is complex. Most patients present the symptoms of disease exacerbation, such as increased expectoration of purulent sputum, dyspnoea, loss of appetite and or increased body temperature. These symptoms are common for all types of infective exacerbations, irrespective of the type of cultured microorganisms.

Chest CT plays an important role in the diagnostic algorithm of both cystic fibrosis and NTMLD. In CF it is used to assess the extent of the disease, complications and response to the implemented treatment [16,17].

CF typically presents on chest CT with bilateral bronchiectasis that dominates in the upper lung lobes, bronchial wall thickening, mucus plugging, emphysema, air trapping, atelectasis, acinar nodules, thickening of interlobular and intralobular septa as well as areas of ground glass opacities [16,18]. It was recommended to perform chest CT biennially in stable CF patients [17].

Bronchiectasis is increasing in number and in size along with disease progression, due to chronic inflammation. Mucus plugging may result in partial or total obstruction of the bronchial lumen. At the late stage of the disease, partial lung cirrhosis is described. Reactive lymphadenopathy is often observed.

NTMLD presents with two types of radiological changes: nodular-bronchiectatic and cavitary [15,19].

The bronchiectatic form was most often diagnosed in non-CF, in middle-aged females with no predisposing factors [20]. Chest CT shows bronchiectasis localized in the middle lobe and lingua, as well as small intra-lobular, poorly defined nodules, which are signs of bronchiolitis.

The cavitary form of NTMLD was mostly observed in older, white males with underlying chronic pulmonary disease [20]. Small, thin-walled cavities are usually localized in the upper lobes. On such occasions, tuberculosis has to be taken into consideration in the differential diagnosis.

The recognition of radiological features of NTMLD in CF patients is extremely difficult. The nodular-bronchiectatic form of NTMLD is practically indistinguishable from the radiologic appearance of CF. Occasionally, new nodular opacities localized in the middle lobe and lingua may be indicative of NTM infection, as was shown in two of our patients.

The cavitary form of NTMLD is easier to recognize in CF patients, as lung infiltrations with cavitation are not typically seen in the course of bacterial infections in CF. Serial analysis of chest CT in a single CF patient may reveal new nodular opacities, not resolving in the course of anti-bacterial therapy. In some of them, cavitation appears in the course of the disease. Such presentation was described in three of our patients. All of the patients with lung cavitation fulfilled the diagnostic criteria of NTMLD. Based on our experience, cavitary lesions on chest CT were the significant predictors of NTMLD not only in CF but also in COPD patients [21].

Cavitary form of NTMLD in CF patients is combined with a worsening in prognosis. Abate et al. found cavitary lesions in 30% of CF patients, this radiological presentation was combined with a three-fold increase in death risk [8].

In the majority of our CF patients with positive NTM isolates, chest CT was not indicative of NTMLD. Therefore, it is of extreme importance to perform periodic sputum cultures for mycobacteria in CF patients. Nick et al. recommended annual NTM screening in CF [6]. At our center, cultures for mycobacteria are performed in every CF patient at least once a year. Bacterial decontamination of sputum samples referred for mycobacterial diagnostics is mandatory. Serial mycobacterial cultures are the principle of screening for NTMLD in CF patients.

Respiratory samples analysis, performed at our center, revealed the presence of NTM in 7% of CF patients. *M. avium* was the dominating type, identified in 55% of patients. *M. avium*/MAC was also the most frequent isolates in the US populations of CF patients [8], whereas *M. abscessus*—in European CF patients [10,11,22].

Increasing the isolation rate of *M. abscessus* in CF centers may indicate in-hospital transmission between the patients or environmental source of acquisition [4,5]. Recent molecular epidemiologic investigations (whole genome sequencing g-WGS) indicate that health-care related transmission of *M. abscessus* is rare [22,23]. Nevertheless, Hassan et al. found that 15% of clustering of MAC isolates concerned patients sharing CF centers, thus indicating a common source of infection [24].

In 2 of our patients (18%), *M. chimaera* was identified. The clinical course of the disease was fulminant in one patient. High virulence of *M. chimaera* was also confirmed by Cohen-Bacrie et al. in an 11-year-old boy with CF [25]. Larcher et al. reported four cases of *M. chimaera* NTMLD in CF, clinical improvement and stabilization of spirometry values were observed in three of them, in the course of treatment, and marked worsening of spirometry in one, that was not treated [12].

In all of our study groups, respiratory specimens were also positive for other bacteria, cultured at the time of NTM isolation. The bacterial cultures of sputum revealed *P. aeruginosa* in 63% of patients and *S. aureus*, in 52%. Such bacterial species are typical for the adult population of CF patients, and they are cultured with increasing frequency at the time of advanced lung disease. In the previous publication from our CF center, concerning 89 CF patients observed between 2008 and 2011, *P. aeruginosa* was cultured in 55.6% of specimens and *S. aureus*—in 37.8% [26].

Fungal infections are diagnosed frequently in CF patients, due to the presence of thick mucus and impaired mechanism of bronchial clearance, the mechanisms that prolong exposition to inhaled spores. The common use of broad-spectrum antibiotics is regarded as a risk factor for yeast infection [27]. A previous study conducted in our center, which concerned 217 respiratory specimens obtained from 42 CF patients, revealed 205 (68%) strains of yeast (mainly *C. albicans*) and 96 (32%) strains of filamentous fungi (mainly *A. fumigatus*) [28].

The most interesting microbiological phenomenon in the patients with CF and in non-CF bronchiectasis is the frequent co-habitation of NTM and fungi. The results of a recently published meta-analysis indicated that the risk of NTM isolation in adult CF patients was increased by 2.75 times in those with *A. fumigatus* colonization, compared to the non-colonized group [29]. It is not clear whether the increased risk of NTM isolation in patients colonized with *A. fumigatus,* depends on direct interactions between both microorganisms. *A. fumigatus* colonization favors T-helper 2 CD4+ T cell response, downregulating the cytokines responsible for NTM eradication [29]. It is also possible that the cohabitation of both pathogens concerns patients with profound lung structural disturbances [10,29]. Additionally, some CF patients colonized with *A. fumigatus*, are recognized with ABPA and treated with oral steroids. ABPA was listed as another possible risk factor for NTM isolation [10,13,30]. In the present study group, 46% of the patients were diagnosed with ABPA, prior to NTM isolation.

The other fungal pathogens frequently isolated in CF patients are members of *Candida* spp. Cuthbertson et al. reported that, based on PCR analysis of the fungal microbiome, *Candida* spp. was found more frequently in CF patients compared to *A. fumigatus*, and that it was one of the risk factors of NTM infection [31]. In the present study, *Candida* spp. was cultured simultaneously with NTM in 82% of patients.

In summary, the diagnosis of NTMLD in CF is very difficult. Most patients present with new clinical signs of infection, but they improve with standard antibacterial therapy. Hemoptysis, which may indicate NTMLD, is frequently observed in CF patients without NTMLD. Weight loss listed as one of the signs of NTM infection can be present in CF due to PEX or malabsorption syndrome. Thus, there are no specific complaints that could indicate NTM infection in CF patients. The early radiologic appearance may be non-specific, as discussed already. Therefore, the role of frequent sputum cultures for NTM and comparison of previous chest CT examinations are of extreme importance.

The treatment of NTMLD in CF patients is complicated, due to many drug interactions and the length of therapy. According to recent publications, CF treatment with CFTR modifiers reduces the frequency of PEX [32]. Preliminary studies indicate also the possibility of a lower risk of NTM infections in patients receiving CFTR modifiers [33].

## Figures and Tables

**Figure 1 diagnostics-12-01514-f001:**
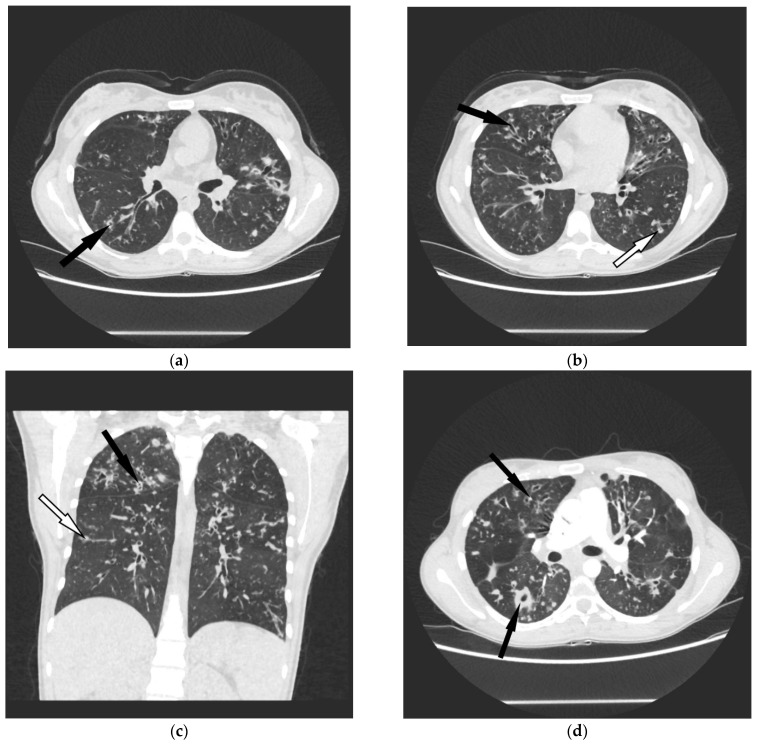
26-years old female with cystic fibrosis and NTMLD (cavitary form). Evolution of chest CT changes from 2018 (**a**–**c**) to 2020 (**d**–**f**). (**a**–**c**) Chest CT scans (lung window, axial and coronal view) demonstrating bronchiectasis of the large and smaller airways (black arrows), centrilobular nodules (white arrows) and bronchial wall thickening predominating in the upper lobes. (**d**–**f**) Chest CT images (lung window, axial and coronal view) demonstrate progression of changes—bronchiectasis has worsened, inflammatory changes became larger, thick wall cavities have appeared (black arrows) which can indicate pulmonary infection caused by NTM.

**Figure 2 diagnostics-12-01514-f002:**
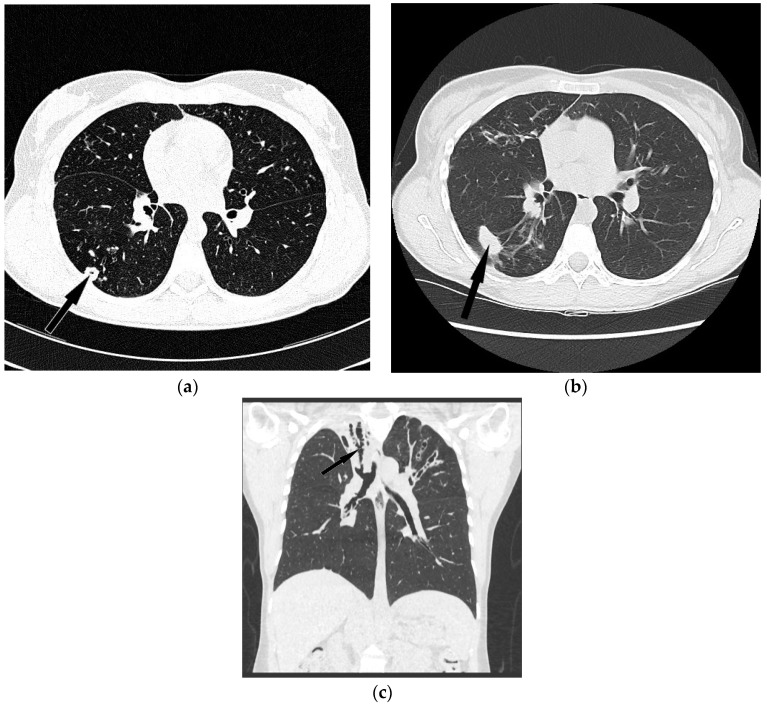
Forty two-year old female with cystic fibrosis and NTMLD (cavitary form). (**a**) CT scan (lung window, axial view) shows thick walled cavity in the 6th segment of right lung (black arrow) which may suggest pulmonary NTM infection. (**b**) CT scan (lung window, axial view) after 6 years in the place of previously visible cavity shows consolidation (black arrow). (**c**) CT scan (lung window, coronal image) presents upper right lobe collapse with cystic bronchiectasis involving the whole bronchial tree (black arrow).

**Table 1 diagnostics-12-01514-t001:** Results of microbiological evaluation in 11 CF patients with positive NTM isolates.

Case N°	NTM Sputum N° pos.	NTM b.wash. N° pos.	Type NTM Isolate	Pseud. Aerug.	Staph. Aureus	Other Bacteria	Asp. Fumig.	Cand. Albic.	Other Fungi
1	3	0	*M. kansasii*	yes	yes	no	no	yes	no
2	2	1	*M. avium*	no	yes	*S. maltoph.*	yes	yes	no
3	0	1	*M. avium*	no	no	*P. fluoresc.*	no	no	no
4	6	1	*M. avium*	no	no	*E. coli, Kl. pneum.* *E. cloacae*	yes	yes	*Penicil.* spp.
5	2	1	*M. avium*	yes	no	no	no	yes	*C. glabr.*
6	4	0	*M. avium*	yes	no	*Achr. xylosox*	yes	yes	*C. glabr.*
7	4	1	*M. abscessus*	yes	yes	no	no	yes	*Penicil.* spp.
8	3	0	*M. lentiflav.*	yes	yes	no	no	yes	no
9	2	1	*M. avium*	no	yes	*S. maltoph.*	yes	yes	*A. flavus, Penicil.* spp.
10	2	1	*M. chimaera*	yes	no	no	yes	yes	no
11	3	0	*M. chimaera*	yes	yes	no	no	no	no

NTM—nontuberculous mycobacteria; Case N°—case number; NTM sputum N° pos.—number of positive NTM sputum specimens; NTM b.wash. N^o^ pos.—number of positive NTM bronchial washing specimens; *Psed. aerug.*—*Pseudomonas aeruginosa*; *Staph. aureus*—*Staphylococcus aureus*; *Asp. fumig.*—*Aspergillus fumigatus*; *Cand. albic.*—*Candida albicans*; *Achr. xylosox.*—*Achromobacter xylosoxidans*; *M. kansasii**—Mycobacterium kansasi*; *M. avium*—*Mycobacterium avium; M. lentiflav.**—Mycobacterium lentiflavum; M. chimaera*—*Mycobacterium chimaera*; *S. maltoph.*—*Stenotrophomonas maltophilia*; *P. fluoresc.*—*Pseudomonas fluorescens*; *E. coli*—*Escherichia coli; Kl. pneum.*—*Klebsiella pneumoniae*; *E. cloacae*—*Enterobacter cloacae*; *Penicil.* spp.—*Penicillium species*; *C. glabr.*—*Candida glabrata*; *A. flavus*—*Aspergillus flavus*.

**Table 2 diagnostics-12-01514-t002:** NTMLD diagnostic criteria based on ATS/IDSA recommendations [15].

Clinical	Pulmonary symptoms, nodular or cavitary opacities on chest radiograph, or a high-resolution CT scan that shows multifocal bronchiectasis with multiple small nodules and 2. Appropriate exclusion of other diagnoses
Microbiologic	Positive culture results from at least two separate expectorated sputum samples or 2. Positive culture results from at least one bronchial wash or lavage or 3. Transbronchial or other lung biopsies with mycobacterial histopathological features (granulomatous inflammation or acid fast bacilli) and positive culture for NTM of lung specimen, bronchial wash, or sputum (at least one)

## Data Availability

Not applicable.

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
