# Peer review of "Nontuberculous Mycobacterial Lung Disease in the Patients with Cystic Fibrosis—A Challenging Diagnostic Problem"

_diagnostics, 2022, doi:10.3390/diagnostics12071514_

Round 1

Reviewer 1 Report

Nontuberculous mycobacterial lung disease in the patients with cystic fibrosis – a challenging diagnostic problem

Dear author and editor:

The author shed a light about the difficulty in CF diagnosis caused by NTM and suggested that the frequent culture sputum and chest CT are fundament in differential diagnosis combined with molecular test.

The article could be published after a minor revision.

I have some comments:

·         Because of NTMs are naturally resistant bacteria so the treatment is also a challenge. The author talked a little about the treatment and the response to the treatment. the author could talked more about this point.

·         Is there any resistant case in the CF patient with a positive NTM isolate.

·         The reasons behind the role of fungal infection in NTM development are not clear in the discussion.

Thank you very much, best regards

Reviewer 2 Report

This case report is well-written and provides details of the patients and disease prognosis. I have the following comments about the article.

  •  The introduction section is small and repetitive to the abstract section. The addition of published literature will provide extra strength to the article.
  • The full form of Mycobacterium is not spelled anywhere before using M. as an abbreviation. Please correct that.
  • A few scientific names need to be italicized. Please see attached file.
  • "The project was accepted by the IEC". Is this the protocol approval from ICE? If yes, please change the statement accordingly.
  • In the result section and other parts of the manuscript while representing the numbers 'period' is replaced with a comma but in some places, it is still a period. Please see the attached file for more details. 
  • Minor typographical errors are there. Please see attached file. 
  • Table 1: Try to adjust table 1 and all heading should be in one line. In the present form, it is difficult to read the column heading and then correlate it with next page values. The same applies to table 2.
  • In fig 1 and 2, highlight the area of interest with an arrow mark for the reader new to the field. 
